# Multimodal Concept Bottleneck Models

## Abstract

Concept Bottleneck Models (CBMs) enhance the interpretability of deep neural networks by aligning the features extracted from images with human-interpretable concepts. However, existing CBMs rely on a linear classifier, which not only restricts them to a fixed set of predefined classes, but also allows the model to bypass concept activations and compensate with non-concept predictive cues. In this paper, we propose Multimodal Concept Bottleneck Model (MM-CBM) to address these issues and extend CBMs into CLIP structure. MM-CBM utilizes dual Concept Bottleneck Layers (CBLs) to align both the image and text embeddings into interpretable features. This allows us to perform new vision tasks like zero-shot classification or image retrieval in an interpretable way. Compared to existing methods, MM-CBM achieves 23.28% accuracy improvement on average across four standard benchmarks. Our method maintains high accuracy, staying within 5% of black-box model performance under finetuned cases while offering greater interpretability.

## 1 Introduction

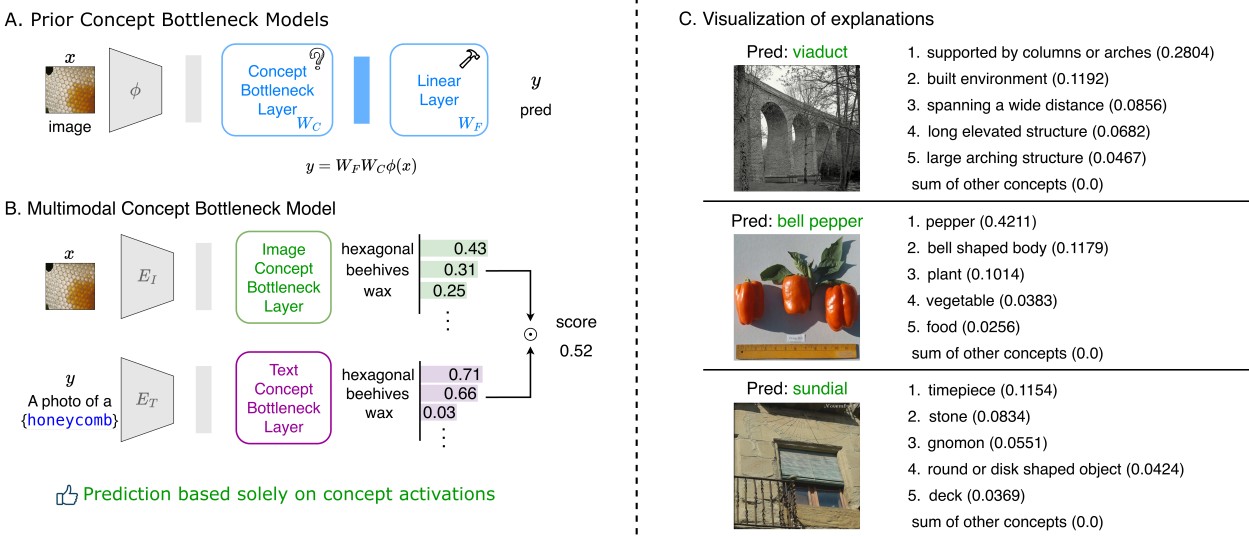

Figure 1: **A.** Previous workflow of concept bottleneck models. **B.** Our MM-CBM makes predictions based solely on concept responses. **C.** Visualization of explanations on several samples.

The opacity and lack of interpretability of deep learning models hinder their deployment in real-world applications. To mitigate this, numerous post-hoc neuron-level explanation methods have been proposed, aiming to understand the semantics of individual neurons (Goh et al., 2021; Oikarinen & Weng, 2024; Bau et al., 2017; Hernandez et al., 2021; Khan et al., 2023; Shaham et al., 2024; Bills et al., 2023). However, these methods often struggle with polysemanticity, where a single neuron encodes multiple, potentially unrelated concepts, limiting their reliability.

Table 1: Comparative analysis of methods based on evaluation, flexibility, and interpretability. Here, ✓ denotes the method satisfies the requirement, △ denotes the method partially satisfies the requirement, and × denotes the method does not satisfy the requirement. We compare with SOTA methods including LF-CBM (Oikarinen et al., 2023), Labo (Yang et al., 2023), LM4CV (Yan et al., 2023) and VLG-CBM (Srivastava et al., 2024).

| Method | Evaluation | | Flexibility | | Interpretability | |
|---|---|---|---|---|---|---|
| | Zero-shot generalization | Sparse explanation | Flexible backbone | Free text input | Free from linear classifier | Multimodal interpretability |
| *Baselines:* | | | | | | |
| CLIP(Radford et al., 2021) | ✓ | × | ✓ | ✓ | ✓ | × |
| LF-CBM(Oikarinen et al., 2023) | × | △ | ✓ | × | × | △ |
| LaBo(Yang et al., 2023) | × | × | × | × | × | △ |
| LM4CV(Yan et al., 2023) | × | × | × | × | × | △ |
| VLG-CBM(Srivastava et al., 2024) | × | ✓ | ✓ | × | × | △ |
| *This work:* | | | | | | |
| MM-CBM | ✓ | ✓ | ✓ | ✓ | ✓ | ✓ |

As an alternative, researchers have developed intrinsically interpretable models such as Concept Bottleneck Models (CBMs) (Koh et al., 2020; Oikarinen et al., 2023; Yan et al., 2023; Srivastava et al., 2024; Yang et al., 2023). CBMs introduce a human-interpretable concept bottleneck layer (CBL) before the classifier, ensuring that predictions are grounded in semantically meaningful concepts. Despite their advantages, existing CBMs rely on a linear classifier placed after the CBL, which introduces two limitations. (1) **Restriction to predefined labels:** Conventional CBMs can only predict among a fixed set of categories and cannot support natural language queries. (2) **Dependence on the classifier:** Recent work (Yan et al., 2023; Srivastava et al., 2024) shows that the downstream linear layer may circumvent concept activations, allowing the model to achieve high accuracy even if the CBL weights are randomly initialized.

On the other hand, powerful vision-language models such as CLIP overcome the limitation of fixed label sets by performing open-vocabulary recognition through image–text embedding similarity. However, the reasoning behind CLIP's decisions remains opaque: the model does not reveal which human-understandable concepts contribute to a high similarity score. This makes it difficult to interpret why a prediction is made or diagnose model failures. In other words, CBMs provide interpretability but lack flexibility, while CLIP is flexible but lacks interpretability. A natural question arises: *Can we design an architecture that supports open-vocabulary predictions while maintaining full transparency in the decision-making process?*

To address this, we propose a new framework called **Multimodal Concept Bottleneck Model (MM-CBM)**. Unlike existing CBMs that rely on a single CBL, MM-CBM introduces **dual CBLs**, one for the image inputs and the other for text input, ensuring that the same conceptual semantics are aligned across both modalities. The text encoder converts arbitrary textual inputs into concept responses, while the image encoder maps visual inputs into the same concept space as shown in Fig. 1B. During inference, predictions are made by computing the similarity between concept responses from both modalities. Additionally, by leveraging the pretrained architectures and knowledge of Vision-Language Models (VLMs), MM-CBMs can achieve performance close to state-of-the-art zero-shot VLMs while providing interpretability.

**Our key contributions are summarized as follows:**

- We propose the first **dual concept bottleneck layers** CBM framework that incorporates concept bottleneck layers across both visual and textual modalities. This design enables complex multimodal tasks such as zero-shot generalization, image retrieval, and open-vocabulary recognition, while preserving concept-level interpretability.

- We establish a **fully transparent reasoning pipeline** in which all final predictions are derived solely from interpretable concept responses. By integrating the concept bottleneck mechanism into a vision-language framework, our method effectively **renders CLIP interpretable**, providing

human-understandable reasoning behind image–text similarity while maintaining open-vocabulary flexibility.

- Extensive experiments demonstrate that our framework significantly outperforms existing CBM methods under the ANEC-5 metric (accuracy under NEC = 5) (Srivastava et al., 2024), achieving **23.28%** average accuracy improvement across four standard benchmarks in the fine-tuned setting. Furthermore, our model attains performance comparable to black-box counterparts, remaining within 5% of their accuracy while offering complete interpretability.

## 2 Related work

**Global neuron-level explanations.** Recent advances in representation-based post-hoc explanation methods have provided new insights into understanding neural network behavior at a global level. Bau et al. (2017) aligned neuron activations with human-labeled image regions using manually annotated datasets, thereby assigning semantic concepts to individual neurons. Kalibhat et al. (2023); Hernandez et al. (2021) identified highly activated image regions through predefined submodules or image captioning models to generate neuron-level explanations. More recently, Oikarinen & Weng (2024; 2023) introduced a concept activation matrix to quantify the similarity between neuron activations and predefined concepts, either through direct computation or predictive modeling, enabling a more structured and scalable interpretation framework. The inherent polysemanticity of neurons in modern neural networks forms the foundation upon which we build our CBL: by linearly combining neuron activations, we are able to synthesize clear, human-aligned concepts.

**Concept bottleneck models (CBMs) for classification.** CBMs (Koh et al., 2020) aim to build intrinsically interpretable models by aligning intermediate representations with human-understandable concepts. A typical CBM consists of two components: a *concept predictor* and a *label predictor*. Given an input $x \in \mathcal{X}$ and a feature extractor $\phi$, the model first maps extracted features $\phi(x)$ to concept activations $c = W_C \phi(x) \in \mathbb{R}^{|C|}$ via a embedding matrix $W_C$, where $C$ denotes the set of candidate concepts. Each dimension in $c$ corresponds to a specific interpretable concept. The final prediction is then obtained by applying a linear classifier parameterized by $W_F$ on top of the concept space. This yields a modified prediction formulation of the form:

$$\hat{y} = W_F W_C \phi(x). \tag{1}$$

This formulation supports *modular reasoning*, enabling inspection, intervention, and editing of intermediate concept activations to enhance interpretability and controllability. Recent works (Oikarinen et al., 2023; Yang et al., 2023; Yan et al., 2023; Hu et al., 2025; Yu et al., 2025; Liu et al., 2025; Zhang et al., 2025) have also explored more flexible training paradigms to improve the generalization and applicability of CBMs, such as leveraging weak supervision, soft labels, or multimodal signals. However, as pointed out by Yan et al. (2023); Srivastava et al. (2024), when the dimensionality of the concept layer is sufficiently large, even randomly embedded features can suffice for a linear classifier to approximate the original prediction. That is, given any projection $W_C$, even a randomly initialized one, it is possible to analytically recover a classifier $W_F$ such that $\hat{y} = W_F W_C \phi(x) \approx W \phi(x)$, where $W$ is the original classifier, as shown in Fig. 1A. This undermines the faithfulness and constraint role of the bottleneck layer. In contrast, our approach incorporates an additional text CBL, enabling responses to arbitrary textual descriptions and generating corresponding class weights, effectively extending concept coverage beyond fixed pre-training categories. A detailed comparison between our method and prior CBMs is provided in Table 1.

**CLIP and its interpretability.** CLIP (Radford et al., 2021) is a large-scale vision-language model trained on extensive image-text pairs using natural language supervision. It achieves strong zero-shot classification performance by encoding both images and text into a shared embedding space and computing similarity scores for prediction. Due to its strong generalization and semantic understanding capabilities, CLIP representations have been widely adopted in tasks such as semantic segmentation, object detection, visual question answering (VQA), and prompt generation for generative models. Numerous variants have been developed to enhance generalization (Sun et al., 2023; Zhai et al., 2023) and computational efficiency (Li et al., 2023).

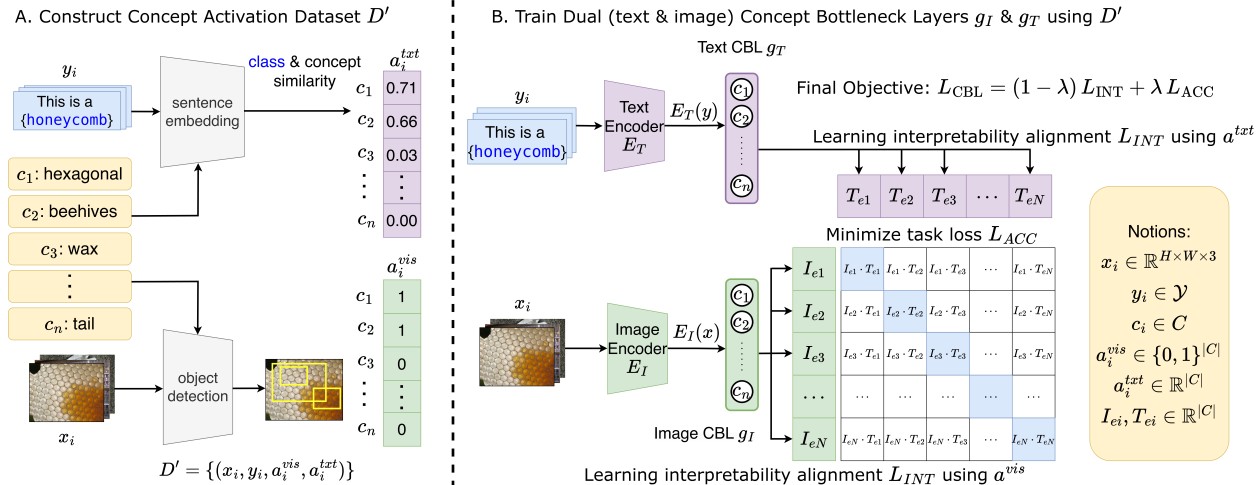

Figure 2: Overview of MM-CBM. (Left) **A.** Extracting high-quality concept annotations for each modality. (Right) **B.** Using an auxiliary dataset to train dual CBLs, jointly optimizing interpretability alignment and task performance.

Several efforts have also been made to interpret CLIP's internal representations. Goh et al. (2021) revealed the presence of multimodal polysemantic neurons within CLIP, showing that individual neurons can encode multiple abstract visual and textual concepts. Bhalla et al. (2024) used dictionary learning to decompose CLIP representations into interpretable semantic components. Menon & Vondrick (2023) proposed a training-free approach that leverages large language models (LLMs) to interpret CLIP's predictions, thereby improving both transparency and performance.

# 3 Method: MM-CBM

In this section, we introduce **Multimodal Concept Bottleneck Models (MM-CBM)**, a novel framework designed to improve the transparency and interpretability of multimodal reasoning by establishing dual Concept Bottleneck Layers (CBLs). Unlike traditional CBMs that rely on a final linear classification head, MM-CBM operates entirely within the concept space, thereby eliminating the dependence on the linear classifier and enabling fully transparent inference.

By incorporating text-based concept encodings, our approach supports a wider variety of natural language inputs, removing the limitation of fixed N-way classification and enabling open-vocabulary image-text matching and zero-shot generalization. MM-CBM is composed of three main stages: (1) **Collecting concept activation data**, (2) **Training dual concept bottleneck layers**, and (3) **Performing inference at test time**.

## 3.1 Collecting concept activation data

Let $E_I : \mathcal{X} \to \mathcal{I}$ denote the image encoder and $E_T : \mathcal{Y} \to \mathcal{T}$ denote the text encoder of CLIP, which map input images and texts into a unified latent representation space. Here, $\mathcal{X} = \mathbb{R}^{H \times W \times 3}$ represents the image space, and $\mathcal{Y}$ denotes the text space. The shared latent space is $\mathcal{I}, \mathcal{T} = \mathbb{R}^d$, where $d$ is the dimensionality. We denote the original dataset used for training CBLs as $D = \{(x_i, y_i)\}$, where $x_i \in \mathcal{X}$ is the $i$-th image, $y_i \in \mathcal{Y}$ is its corresponding textual label. For a fixed-label classification task, let $Y$ be the set of all possible class labels. The label index of $y_i$ is denoted by $l_i \in \{0, 1, \ldots, |Y| - 1\}$, such that $Y_{l_i} = y_i$.

**Concept set generation.** Following recent works (Oikarinen et al., 2023; Yang et al., 2023; Yan et al., 2023), we adopt a fully automated pipeline that queries an LLM for each class label $y \in Y$ to generate a candidate concept set $C_y$, with final concept set $C = \bigcup_{y \in Y} C_y$, reducing annotation cost and avoiding reliance

on scarce datasets with human-defined concepts. For zero-shot classification, we leverage the task-agnostic concept set from SpLiCE (Bhalla et al., 2024).

**Collecting concept labels.** With the candidate concept set $C$ in place, we construct a concept activation dataset $D' = \{(x_i, y_i, a_i^{vis}, a_i^{txt})\}$ by augmenting each image-text pair with two additional labels as shown in Fig. 2A (left):

- $a_i^{vis} \in \{0, 1\}^{|C|}$, a binary vector indicating which concepts appear in the image $x_i$,

- $a_i^{txt} \in \mathbb{R}^{|C|}$, quantifies how strongly each concept in $C$ relates to the text label $y_i$.

To equip the model with interpretable supervision, we first extract binary concept labels for each image based on OWLv2's open-source object detection results (Minderer et al., 2024). The image concept label $[a_i^{vis}]_j$ for concept $c_j$ is defined as:

$$[a_i^{vis}]_j = \begin{cases} 1, & \text{if concept } c_j \text{ appears in image } x_i, \\ 0, & \text{otherwise.} \end{cases} \tag{2}$$

For each training image $x_i$, we prompt OWLv2 with the class-specific concept set $C_{y_i}$. The model predicts a set of bounding boxes $B = \{(b, f, c)\}$, where $b$ represents the box coordinates, $f$ is the confidence score, and $c \in C_{y_i}$ is the detected concept. If the confidence $f$ exceeds a predefined threshold $T$, we consider the concept $c$ to be present in image $x_i$.

To compute $a_i^{txt}$, the text-concept similarity vector, we define each entry $[a_i^{txt}]_j$ as the semantic similarity between the text label $y_i$ and concept $c_j$:

$$[a_i^{txt}]_j = \text{sim}(y_i, c_j). \tag{3}$$

For the similarity function, we follow the Automatic Concept Scoring (ACS) method from CB-LLM (Sun et al., 2025), where similarity is defined as:

$$\text{sim}(y_i, c_j) = \mathcal{E}(y_i) \cdot \mathcal{E}(c_j), \tag{4}$$

with $\mathcal{E}(\cdot)$ denoting the text embedding generated by a language model. In our implementation, we use the `all-mpnet-base-v2` model (Wolf et al., 2020) as the text encoder.

### 3.2 Training dual concept bottleneck layers

Given the concept activation dataset $D'$, we train a pair of Concept Bottleneck Layers (CBLs): one for identifying concept presence in images, and the other for capturing the association between concepts and textual labels. Our training objective consists of two components: an **interpretability loss** $L_{\text{INT}}$ and a **classification loss** $L_{\text{ACC}}$ as shown in Fig. 2B (right).

**Interpretability loss $L_{\text{INT}}$.** To explicitly align the outputs of the concept bottleneck layers (CBLs) with human-interpretable concepts, we define an interpretability loss that supervises both the image and text sides using binary and soft labels, respectively. Let $g_I : \mathcal{I} \to \mathbb{R}^{|C|}$ and $g_T : \mathcal{T} \to \mathbb{R}^{|C|}$ denote the image and text CBLs, respectively, where $|C|$ is the number of concepts. These CBLs project image and text features into a shared concept space. To enforce consistency between predicted concept activations and ground-truth annotations in $D'$, we define the interpretability loss as:

$$L_{\text{INT}} = \frac{1}{|D'|} \sum_{i=1}^{|D'|} L_I(\hat{a}_i^{vis}, a_i^{vis}) + L_T(\hat{a}_i^{txt}, a_i^{txt}), \tag{5}$$

where $\hat{a}_i^{vis} = g_I \circ E_I(x_i), \quad \hat{a}_i^{txt} = g_T \circ E_T(y_i)$ are the predicted concept activations for the image $x_i$ and text $y_i$, respectively. Here, $E_I$ and $E_T$ are the image and text encoders introduced in Section 3.1, and

$a_i^{vis}, a_i^{txt} \in \mathbb{R}^{|C|}$ are the concept label vectors for image and text respectively. We adopt binary cross-entropy (BCE) for the image-side loss:

$$L_I\big(\hat{a}_i^{vis}, a_i^{vis}\big) = -\sum_{c=1}^{|C|} a_{i,c}^{vis} \log \hat{a}_{i,c}^{vis} + (1 - a_{i,c}^{vis}) \log(1 - \hat{a}_{i,c}^{vis}), \tag{6}$$

and negative cosine-cubed similarity for the text-side loss:

$$L_T\big(\hat{a}_i^{txt}, a_i^{txt}\big) = -\frac{(\hat{a}_i^{txt})^3 \cdot (a_i^{txt})^3}{\big\|(\hat{a}_i^{txt})^3\big\|_2 \, \big\|(a_i^{txt})^3\big\|_2}, \tag{7}$$

reflecting the categorical and continuous nature of the constructed concept labels.

**Task loss $L_{\text{ACC}}$.** To preserve discriminative performance on downstream tasks, we introduce a task-specific classification loss based on the representations in the concept space. Let $I_e = g_I \circ E_I(x)$ and $T_e = g_T \circ E_T(y)$ denote the image and text representations in the concept space.

To ensure interpretability and promote sparsity, i.e., encouraging the prediction to rely on a small subset of semantically meaningful concepts, we draw inspiration from the *number of effective concepts* (NEC) (Srivastava et al., 2024). Specifically, the similarity between $I_e$ and $T_e$ is computed as the sum of the top-$n$ responding dimensions over element-wise products. To further enhance interpretability and suppress the influence of negatively activated concepts, we set all negative elements in the concept vectors to zero before computing the similarity: $I_e^+ = \text{ReLU}(I_e)$ and $T_e^+ = \text{ReLU}(T_e)$. More details are provided in Appendix A.4. The semantic consistency between the image and text is then defined as:

$$z_j = \left( \frac{\sum \text{top-}n(I_e^+ \odot T_{ej}^+)}{\|I_e^+\|_2 \|T_{ej}^+\|_2} \right) \times \exp(\tau), \tag{8}$$

where $z_j$ denotes the similarity score (logits) of the image $x$ with respect to the class or query $y_j$, $\odot$ represents element-wise multiplication. In this case, the classification loss is expressed as:

$$L_{\text{ACC}} = L_{\text{CE}}(z, \ l), \tag{9}$$

where $L_{\text{CE}}$ denotes the cross-entropy loss, and $l$ is the index of the ground-truth label.

**Final objective** To jointly optimize interpretability and task performance, we integrate the interpretability loss and discriminative loss into a unified objective. Notably, our model can also be trained without ground-truth labels, achieving classification accuracy comparable to CLIP; see Appendix A.5 for details. This combined loss function enables the model to learn concept-aligned representations while maintaining strong classification performance, thereby mitigating the risk of the linear layer overfitting to the task and compensating for a poorly interpretable concept space:

$$L_{\text{CBL}} = (1 - \lambda) \, L_{\text{INT}} + \lambda \, L_{\text{ACC}}, \tag{10}$$

where $\lambda \in [0, 1]$ controls the trade-off between interpretability and task accuracy.

### 3.3  Performing inference at test time

During inference, given any image $x_i$ and text $y$, the model outputs two modality-specific concept embeddings: an image concept embedding $I_e = (c_{i1}, c_{i2}, \cdots, c_{im})$ and a text concept embedding $T_e = (c_{t1}, c_{t2}, \cdots, c_{tm})$, where each element $c_j$ reflects the degree to which the $j$-th concept is present in the image or related to the text, $m = |C|$ is the total number of candidate concepts. Semantic consistency between the two embeddings is computed using Eq. 8. The resulting similarity score serves as the class prediction, indicating how well the image matches the queried textual description within the shared concept space. The image–text pair with the highest similarity score is taken as the model's prediction.

Furthermore, because both embeddings are represented in human-interpretable concept dimensions, each prediction can be explicitly explained by inspecting the concept activations that most strongly contribute to

the similarity score. In contrast to traditional CBMs that rely on a linear classifier to associate concepts with categorical labels, our text CBL directly generates concept activations from natural language descriptions. This not only simplifies the inference pipeline but also enables flexible support for open-vocabulary text inputs, making the entire reasoning process of MM-CBM fully transparent, as illustrated in Fig. 2B.

## 4 Experiment

Section 4.1 describes the experimental setup. Section 4.2 compares MM-CBM to existing CBMs and the CLIP backbone under both fine-tuned and zero-shot settings. Section 4.3 examines the impact of the interpretability enhancement techniques introduced in Section 3.2 and Appendix A.4. Section 4.4 investigates how manual interventions can be used to correct certain misclassifications and trace the underlying causes. Section 4.5 presents quantitative interpretability evaluations through human and VLM-based assessments.

### 4.1 Experimental setup

**Datasets**: We conduct experiments on seven datasets spanning different types of visual recognition tasks: (1) **General image classification**: CIFAR-10, CIFAR-100 (Krizhevsky et al., 2009), and ImageNet (Deng et al., 2009); (2) **Fine-grained classification**: Food-101 (Food) (Bossard et al., 2014), CUB (Wah et al., 2011), and Oxford-IIIT Pets (OxfordPets) (Parkhi et al., 2012); (3) **Texture classification**: Describable Textures Dataset (DTD) (Cimpoi et al., 2014). Additionally, we trained MM-CBM on multimodal dataset CC12M (Changpinyo et al., 2021) to test the generalization ability. We follow the standard train/test splits, as detailed in Appendix A.6, and use classification accuracy as the evaluation metric.

**Baselines**: We compare MM-CBM with four interpretable baselines: LF-CBM (Oikarinen et al., 2023), LaBo (Yang et al., 2023), LM4CV (Yan et al., 2023), and VLG-CBM (Srivastava et al., 2024). We also compare against the CLIP backbone under both zero-shot and fine-tune settings.

**Implementation**: We use `gpt-3.5-turbo-instruct` to generate candidate concept sets for datasets. Unless otherwise specified, the trade-off parameter between interpretability and task performance is set to $\lambda = 0.2$, and the NEC is fixed at 5. For fair comparison with prior CBMs, we use CLIP-RN50 as the backbone. All other evaluations are conducted using models trained with CLIP-ViT-L/14. We use the Adam optimizer (Kingma & Ba, 2014) during training. For each batch, we randomly select one sentence from those generated by VLMs as the text input for each category.

In the **fine-tune scenario**, where target datasets and ground-truth labels are available for better adaptation, we compare MM-CBM to CLIP-ViT-L/14 with linear probing. In the **zero-shot scenario**, where the model has never observed any data or labels during training, we compare with the zero-shot performance of CLIP. To accelerate training and reduce computational overhead under the zero-shot scenario, we omit the NEC constraint and directly use the inner product $I_e^+ \odot T_e^+$ instead of the top-$n$ summation.

Table 2: Comparison with other CBMs on ANEC-5 using CLIP RN50. Best results for each benchmark are in **bold**; second-best are underlined.

| Method | Dataset | | | | |
|---|---|---|---|---|---|
| ANEC=5 | CIFAR10 | CIFAR100 | CUB | ImageNet | Average |
| LF-CBM(Oikarinen et al., 2023) | 84.05 | 56.52 | 31.35 | 52.88 | 56.20 |
| LM4CV(Yan et al., 2023) | 53.72 | 14.64 | 3.63 | 3.77 | 18.94 |
| LaBo(Yang et al., 2023) | 78.69 | 44.82 | 41.97 | 24.27 | 47.44 |
| VLG-CBM(Srivastava et al., 2024) | 88.55 | 65.73 | **60.38** | 59.74 | 68.60 |
| **MM-CBM(Ours)** | **88.69** | **66.34** | 58.79 | **70.49** | **71.08** |

Table 3: Test accuracy comparison with black-box CLIP on zero-shot and finetuned settings.

| Method | Dataset | | | | | | |
|---|---|---|---|---|---|---|---|
| | CIFAR-10 | CIFAR-100 | CUB | Food | OxfordPets | DTD | ImageNet |
| *Zero-shot* | | | | | | | |
| CLIP ViT-L/14 | 96.2 | 77.9 | 62.3 | 92.9 | 93.5 | 55.3 | 75.3 |
| **MM-CBM(Ours)** | 95.3 | 75.7 | 36.3 | 84.3 | 76.6 | 50.2 | 66.1 |
| *Finetuned* | | | | | | | |
| CLIP linear probe | 98.0 | 87.5 | 84.5 | 95.2 | 95.1 | 82.1 | 83.9 |
| **MM-CBM(Ours)** | 97.5 | 84.1 | 75.0 | 93.4 | 91.2 | 75.6 | 81.9 |

Table 4: Zero-shot retrieval comparison with black-box CLIP.

| | Text Retrieval | | | | | | Image Retrieval | | | | | |
|---|---|---|---|---|---|---|---|---|---|---|---|---|
| | Flickr30k | | | MSCOCO | | | Flickr30k | | | MSCOCO | | |
| | R@1 | R@5 | R@10 | R@1 | R@5 | R@10 | R@1 | R@5 | R@10 | R@1 | R@5 | R@10 |
| CLIP | 85.3 | 97.4 | 99.2 | 56.3 | 79.3 | 86.6 | 64.9 | 87.2 | 92.0 | 36.6 | 61.1 | 71.1 |
| MM-CBM | 71.7 | 92.5 | 96.8 | 41.9 | 67.8 | 78.1 | 61.8 | 85.1 | 90.8 | 32.9 | 58.5 | 68.9 |

## 4.2 Results

**Comparison with existing CBMs:** Table 2 shows accuracy under NEC = 5 (ANEC-5). MM-CBM achieves performance comparable to the strongest baseline, VLG-CBM, and surpasses others by over 10% accuracy on ImageNet. This suggests that MM-CBM benefits from the rich semantic knowledge embedded in the CLIP backbone, particularly on large-scale datasets. Additional results compared with other CBMs under the zero-shot setting are provided in the Appendix A.6.

**Comparison with CLIP backbone:** Table 3 compares MM-CBM with CLIP under both fine-tuned and zero-shot settings. Across seven datasets, MM-CBM achieves performance comparable to both CLIP's linear-probe and zero-shot settings. A noticeable drop appears on the CUB dataset, particularly in the zero-shot scenario, which we attribute to the limited coverage of fine-grained bird attributes in the candidate concept set. Nevertheless, the performance remains non-trivial, indicating that MM-CBM can still generalize reasonably well even under challenging and attribute-rich conditions.

Furthermore, the image retrieval results in Table 4 demonstrate that MM-CBM preserves task effectiveness while offering explicit, concept-level interpretability. The performance gap relative to black-box CLIP remains under 5%, suggesting that MM-CBM retains most of CLIP's retrieval capability while providing transparent and human-understandable reasoning.

## 4.3 Ablation study

We assess the effect of the non-negative concept space introduced in Section 3.2 and Appendix A.4. We compare our method against alternatives including sigmoid activation, squaring activations, and a baseline without non-negative processing. Concept sparsity is quantified using the $\ell_1$ norm.

For a non-negative vector $v \in \mathbb{R}^{|S|}$ with $|v|_2 = 1$, we have $1 \leq \|v\|_1 \leq \sqrt{|S|}$. Lower $\|v\|_1$ implies higher sparsity, which improves interpretability.

We report the average $\ell_1$ norm of visual ($I_e$) and textual ($T_e$) activations, and average alignment score across validation samples. Table 5 shows that our approach yields at least 5× smaller $\ell_1$ norm for visual activations and 2× smaller for text, compared to other methods. Our alignment score also improves by 3×, suggesting higher prediction confidence. Importantly, increased sparsity does not degrade accuracy but enhances reliability. Additionally, since visual supervision uses binary targets and text uses real-valued

similarity scores, visual concept activations are expected to be sparser. Our method preserves this property, while others reverse it, potentially introducing redundant activations.

Table 5: Ablation study of non-negative setting. Visual and language correspond to the average $\ell_1$ norm of image and text concept activation; Score means the average highest alignment score, the image and prediction alignment score.

| Function | Visual activation $\ell_1$ ↓ | Language activation $\ell_1$ ↓ | Alignment score ↑ |
|---|---|---|---|
| Sigmoid | 11.690 | 11.699 | 0.079 |
| $x^2$ | 58.848 | 23.524 | 0.044 |
| None | 63.030 | 60.665 | 0.002 |
| **ReLU** | **2.703** | **5.668** | **0.271** |

## 4.4 Human Intervention

We further analyze the model's decision process and demonstrate how manual adjustments based on expert knowledge can improve predictions, inspired by Oikarinen et al. (2023). Fig. 3 shows a misclassification where the model predicts "barbershop" due to a strong activation of the concept "a sign that says barbershop," which is not visually present. This can be corrected by setting $T_{e[\text{pred, concept}]} = 0$.

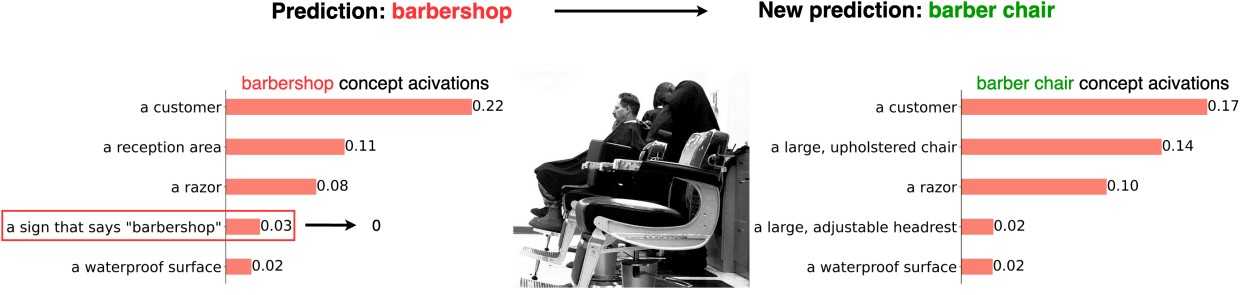

Figure 3: A sample of correcting model prediction by deleting the wrong concept.

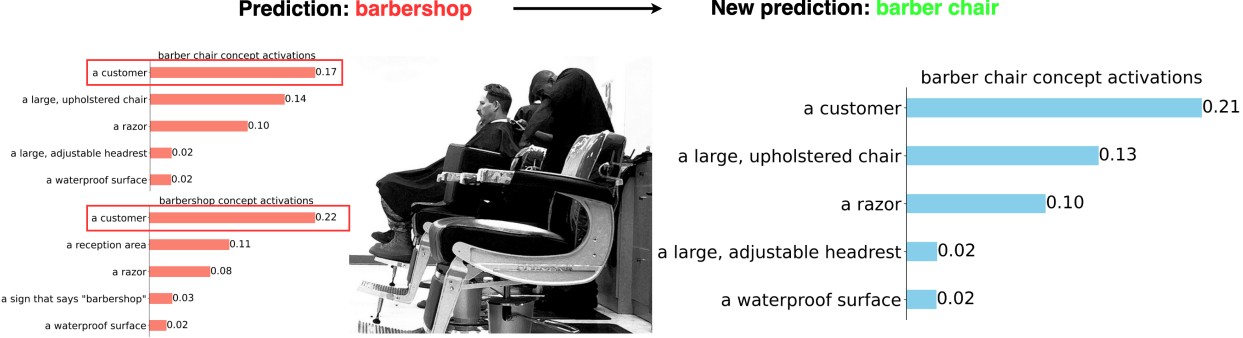

Figure 4: A sample of correcting model prediction by setting the common concept to the same value

Another error occurs when "barbershop" is incorrectly favored due to a higher response to "a customer," despite "barber chair" being the correct label. Equalizing the activation between both classes for that concept ($T_{e[\text{gt, concept}]} = T_{e[\text{pred, concept}]}$) corrects 5 predictions and introduces 2 new errors, shifting predictions from "barbershop" to "barber chair." This leads to a 3% accuracy improvement in a 100-sample subset. Such errors stem from **response bias on shared concepts**, which hinders fine-grained classification when dominant but insufficient features overshadow more specific ones.

The source of this error is traceable: since $I_e$ is identical, the difference lies in $T_e$. Using the label generation model `all-mpnet-base-v2` (Wolf et al., 2020), we find that the similarity score between "barbershop" and "a customer" is 0.3618, compared to 0.3070 for "barber chair." This discrepancy reflects a language-model-induced bias during training.

The Fig. 4 result of manually editing the text concept activation $T_{e[\text{gt, concept}]} = T_{e[\text{pred, concept}]}$ for the case in Fig. 3

## 4.5 Interpretability result

**Human study.** To evaluate the interpretability of our model, we conducted a user study following a similar setting to Label-free CBM (Oikarinen et al., 2023). We compared two variants of MM-CBM (zero-shot and fine-tuned) against SpLiCE (Bhalla et al., 2024). For each method, participants were shown 100 randomly sampled ImageNet images with the correct label. For every sample, we displayed the top five contributing concepts and their corresponding weights, and asked users to rate "which explanation is better" on a Likert scale from 1 to 5.

Across both evaluation settings, users consistently preferred the explanations produced by MM-CBM. Participants further provided qualitative feedback: in the zero-shot setting, MM-CBM offered descriptions that were more relevant to the image in 55.33% of cases and provided more informative explanations in 56.25% of cases. In the fine-tuned setting, these preferences increased to 64.52% and 66.57%, respectively, indicating that MM-CBM delivers more faithful and more informative explanations. Our study was deemed Exempt from IRB approval by the IRB review board at our institution.

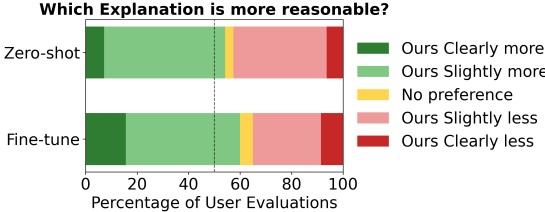

Figure 5: Interpretability result

**Comparison with VLMs** Vision-Language Models (VLMs) are highly capable of understanding the overall semantics of images and generating natural language explanations. This appears similar to the goal of CBMs, so we directly compared explanations from our MM-CBM (ImageNet) with those from VLMs.

Specifically, we prompted each model with the template: "Why is this image categorized as {cls}", and collected 5,000 explanation pairs from imagenet dataset. To evaluate which explanation contained more informative visual concepts, we leveraged the VQA capabilities of VLMs themselves by asking: "Which description has more informative visual concepts in this image?" Notably, to reduce model-specific bias and avoid self-preference in scoring, we separated the roles of evaluator and competitor across models, using different VLMs to act as the "judge" and the "explainer." In our experiments, we used `LLaVA-v1.5-7B` (Liu et al., 2023) and `Llama-3.2-11B-Vision-Instruct` (Touvron et al., 2023), alternating their roles to ensure fairness and robustness. MM-CBM explanations were preferred over LLaVA 1.5 in **4,433** (88.7%) out of 5,000 cases, and over Llama 3.2 in **3,292** (65.8%) cases. These findings suggest that although VLMs are adept at generating high-level semantic interpretations, they tend to overlook fine-grained visual concepts that are central to CBM-style interpretability.

# 5 Case study: image retrieval

| CLIP retrival | MM-CBM explainable results | | |
|---|---|---|---|
| **Level-1** · Query: Give me a crane

This image is a crane because it has:
1. *cranes (0.0468)*
2. *construction site (0.0189)*
3. *model toy (0.0104)* | **The concepts most related to query:**
crane (0.1774)
cranes (0.1377)
heavy equipment (0.0931)
forklift (0.0829) | **The concepts most related to the image**
cranes (0.3398)
construction site (0.3145)
background design (0.2642)
model toy (0.2448) | |
| **Level-2** · Query: Spotted fur

This image contains spotted fur because it has:
1. *fur collar (0.0306)*
2. *fur vest (0.0292)*
3. *leopard (0.0217)* | fur coat (0.1398)
fur collar (0.1172)
faux fur (0.1154)
fur trim (0.1063)
leopard print (0.0908) | image (0.3658)
zara coat (0.3136)
fur vest (0.2950)
leopard (0.2780)
fur collar (0.2613) | |
| **Level-3** · Query: The image shows crane, the animal

This image is an animal crane because it has:
1. *cranes (0.0489)*
2. *wild animals (0.0215)*
3. *lobster (0.0142)* | crane (0.1497)
cranes (0.1228)
wildlife (0.1130)
birds vector (0.0940)
wild animals (0.0904) | cranes (0.3980)
lobster (0.2801)
wild animals (0.2374)
camels (0.2290)
vector format (0.2243) | |
| **Level-4** · Query: This image depicts something broken

This image contains a broken corner because it has:
1. *landfill (0.0242)*
2. *car crash (0.0147)*
3. *image (0.0121)* | car crash (0.0926)
broken heart (0.0842)
conceptual image (0.0838)
landfill (0.0829)
concept illustration (0.0768) | landfill (0.2916)
usb port (0.2101)
vintage camera (0.2067)
footwear (0.1920)
image (0.1901) | |
| **Level-5** · Query: Hit the sack

This shows a sleep-related idiom because it has:
1. toddler bed *(0.0316)*
2. *infant (0.0176)*
3. *blankets (0.0099)* | nap (0.1192)
sleeping (0.1055)
tired (0.0943)
aslleep (0.0931)
toddler bed (0.0871) | toddler bed (0.3628)
infant (0.3456)
london (0.1928)
self portrait (0.1809)
lyrics (0.1805) | |

Figure 6: Image Retrieval on five different levels of queries. By incorporating the textual CBL, the model can infer contextual meanings: in Level-3, the query "crane the animal" successfully retrieves the bird instead of the mechanical object; in Level-4, the term "broken" evokes semantically related scenes; and in Level-5, the model correctly distinguishes between the literal and figurative meanings of polysemous phrases. For better readability, we present the top-3 concepts most relevant to image retrieval, as well as the top-5 activated concepts for each modality. All concepts are ranked based on their original activations.

We replace the fixed linear classifier with a text encoder, enabling flexible and unrestricted text inputs. In this section, we evaluate our model through an image retrieval task: given arbitrary text, the model selects the image with the highest alignment score. This evaluation assesses the model's semantic consistency and generalization ability. To systematically analyze interpretable reasoning in image retrieval, we define five levels of textual queries:

- **Level-1: Ground-truth label queries** – Direct retrieval using exact class labels (e.g., *crane*, *popsicle*, *uniform*).

- **Level-2: Concept-based queries** – Retrieval based on key concepts (e.g., *striped fur*, *spotted fur*, *uniformed fur*), allowing us to test fine-grained concept understanding.

- **Level-3: Hybrid label–concept queries** – Queries that combine class labels and specific concepts (e.g., *crane the animal*, which resolves the ambiguity of the polysemous word "crane," or *cat with striped fur*, which resembles a *tiger*).

- **Level-4: Out-of-distribution queries** – Texts containing unseen labels or novel concepts not present in the training set (e.g., *broken*, which typically describes a state rather than a class in standard classification tasks).

- **Level-5: Polysemous or abstract queries** – Phrases involving semantic ambiguity, idiomatic expressions, or abstract meanings. (e.g., *hit the sack* meaning "go to sleep", *Break a leg*, which literally means "injure a leg" but figuratively conveys "good luck" in theatrical slang

This hierarchical evaluation setup helps validate the model's semantic alignment and its ability to generalize beyond predefined labels or fixed concept sets. We conduct all experiments using the model trained on CC12M, and representative examples for each query level are shown in Fig. 6. The retrieval results demonstrate that our model's semantic understanding is largely consistent with both human judgment and the backbone model, while additionally providing an interpretable account of how such semantic decisions are formed.

We further present additional cases in the Appendix A.9 Fig. A.4, particularly for Level-4 and Level-5 queries, where the showcases are less accurate or even incorrect compared to the original black-box model. In these cases, the text modality often captures concepts that are semantically similar to the query. However, since our visual concept labels are constructed via object detection, the visual branch is inherently less sensitive to abstract semantics and instead responds more strongly to concrete objects. Moreover, as shown in Table 5, visual activations are generally higher than textual ones, leading the final explanations to be dominated by visual signals. This imbalance helps explain the observed failure cases. In contrast, for more concrete queries (Level-1 to Level-3), concept activations are consistently stronger, indicating that the model is more confident when handling explicit and visually grounded semantics.

## 6 Conclusion

In summary, we present MM-CBM, a simple yet effective framework that turns vision–language foundation models into interpretable, concept-based predictors. By extracting aligned concepts from both image and text encoders, MM-CBM provides a transparent alternative to CLIP, preserving its flexibility while offering explicit, human-understandable reasoning, and achieves performance competitive with both traditional CBMs and black-box VLMs. Our use-case studies show that MM-CBM supports rich semantic reasoning, accurately resolves contextual and polysemous queries in image retrieval, and delivers explanations that users consistently judge as more faithful and more informative than prior methods. These results highlight the practical value of MM-CBM and suggest its potential as a general, trustworthy building block for multimodal applications such as retrieval, captioning, and visual question answering.

## Broader Impacts

MM-CBM improves the interpretability of vision–language foundation models by grounding predictions in human-understandable concepts. However, it may be subject to misuse, whereby harmful or discriminatory concepts are injected during the training process, potentially leading to biased or undesirable outputs. In addition, since the training data is labeled based on black-box foundation models, the learned concepts may not fully align with human understanding, introducing further risks. To mitigate these issues, MM-CBM enables human intervention during inference and supports error tracing, facilitating more transparent diagnosis and partial control over model behavior.

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

# A Appendix

## A.1 Overview

The appendix covers: A.2 problem formulation; A.3 concept set generation; A.4 interpretability enhancement strategies; A.5 unsupervised adaptation via knowledge distillation; A.6 additional experiments; A.7 additional ablation studies; A.8 alternative backbones; and A.9 image retrieval examples.

## A.2 Problem formulation

Given an input image $I$ and a text description $T$, we aim to estimate the conditional probability $p(T|I)$, i.e., the likelihood that $T$ correctly describes $I$. We assume that both modalities share an intermediate concept space $c_i{}_{i=1}^{K}$, where each concept $c_i$ captures a semantic attribute relevant to both image and text. Strict equality holds only if the concept set $C$ fully covers the semantic space and the concepts are mutually disjoint; in practice, we adopt this as an approximation.

Under this assumption, the image-text relation can be decomposed through the latent concepts as follows:

$$p(T|I) \approx \sum_i p(T, c_i|I). \tag{A.1}$$

By applying the product rule of probability, we have

$$\begin{cases} p(T, c_i|I) = p(T|c_i, I)\, p(c_i|I), \\ p(T|c_i, I) = p(T|c_i), \end{cases} \tag{A.2}$$

where the second equality assumes that, given the concept $c_i$, the text $T$ is conditionally independent of the image $I$. Substituting into the previous equation yields

$$p(T|I) = \sum_i p(T|c_i)\, p(c_i|I). \tag{A.3}$$

Next, by applying Bayes' theorem, the term $p(T|c_i)$ can be expressed as

$$p(T|c_i) = \frac{p(c_i|T)\, p(T)}{p(c_i)}. \tag{A.4}$$

The concept prior $p(c_i)$ can be marginalized over the image distribution:

$$p(c_i) = \sum_j p(c_i|I_j)\, p(I_j). \tag{A.5}$$

Combining the above relations gives

$$p(T|I) = \sum_i p(c_i|I)\, \frac{p(c_i|T)\, p(T)}{\sum_j p(c_i|I_j)\, p(I_j)}. \tag{A.6}$$

When assuming uniform priors for $p(I)$ and $p(T)$, the proportional relation becomes

$$p(T|I) \propto \sum_i p(c_i|I)\, \frac{p(c_i|T)}{\sum_j p(c_i|I_j)}. \tag{A.7}$$

This formulation provides a probabilistic bridge between the image and text modalities through their shared concept distributions $p(c_i|I)$ and $p(c_i|T)$, which can be directly predicted by our CBL layers, allowing us to evaluate the semantic consistency between $I$ and $T$ via their inferred concept activations. Our work presents an initial implementation of this modeling approach. Future research could further enhance interpretability and task performance by improving the quality of the concept set, e.g., expanding its semantic coverage and reducing overlap, and by refining the conditional concept distribution model to better approximate the true underlying distribution.

### A.3 Concept set generation

Let $C$ denote a fine-grained concept set that semantically explains the images and their corresponding labels in $D$. Such a set can be manually curated by domain experts or automatically generated using large language models (LLMs) (Touvron et al., 2023; Brown et al., 2020). Following recent studies (Oikarinen et al., 2023; Yang et al., 2023; Yan et al., 2023), we adopt a fully automated approach in which, for each class label $y \in Y$, an LLM is queried to produce a candidate concept set $C_y$. Under the label-free CBM setting (Oikarinen et al., 2023), the LLM is prompted as follows:

- List the most important features for recognizing something as a {class}:

- List the things most commonly seen around a {class}:

- Give superclasses for the word {class}:

Here, {class} refers to the class name in the target classification task. The final concept set is obtained as the union of all class-specific sets:

$$C = \bigcup_{y \in Y} C_y.$$

We further refine $C$ using the filtering strategy proposed in label-free CBM (Oikarinen et al., 2023), with the following steps:

1. **Concept length:** Discard concepts exceeding 30 characters to maintain simplicity and interpretability.

2. **Similarity to target classes:** Remove concepts overly similar to target class names, as they undermine the explanatory role of the CBM. Similarity is measured via cosine similarity in a joint text embedding space, combining features from the CLIP ViT-B/16 text encoder and the all-mpnet-base-v2 sentence encoder. Concepts with similarity greater than 0.85 to any target class are excluded.

3. **Redundancy removal:** Eliminate duplicate or near-synonymous concepts to ensure diversity in the bottleneck layer. Using the same embedding space, any concept with cosine similarity above 0.9 to an already retained concept is removed.

This automated generation and filtering process substantially reduces the reliance on manual annotation while enabling scalable construction of rich concept sets, even for datasets lacking human-defined concept annotations.

**Reproducibility.** The quality and diversity of automatically generated concepts directly influence the representational capacity of the semantic space, and consequently affect downstream classification and retrieval performance. However, the stochastic nature of LLMs introduces potential variability in concept generation, raising concerns about stability and reproducibility.

To mitigate this issue, we adopt a controlled and largely deterministic generation pipeline, including fixed prompts, consistent few-shot exemplars, and standardized post-processing steps (deduplication and threshold-based filtering).

We empirically evaluate the stability of this pipeline by repeating the concept generation process five times on CIFAR-100. The resulting concept sets achieve an average pairwise Jaccard similarity of 0.48. Notably, the observed variation is primarily attributable to differences in lexical choice (e.g., synonyms) and granularity, rather than semantic drift. This suggests that the generated concept sets remain semantically consistent across runs, supporting the robustness and reproducibility of our automatic concept generation procedure.

### A.4 Strategies to enhance interpretability

In this section, we introduce three strategies designed to enhance the interpretability of our multimodal CBM model.

**Generation of rich textual information.** In many vision-language datasets, there exists a significant imbalance between the number of images and the granularity of their associated textual labels, where hundreds or even thousands of images may share the same class name. Repeatedly using identical textual inputs during training can introduce undesirable biases and restrict model generalization. To address this issue, we leverage the capabilities of the state-of-the-art multimodal large language model `Llama 3.2-Vision` to generate diverse, semantically rich label descriptions. Specifically, we prompt the model with the following template: *"If I had to describe this image using only one sentence with the words class, it would be: "* For each class label, we randomly select images belonging to that class and generate at least 50 unique textual descriptions. During training, one of these alternative descriptions is randomly sampled for each iteration, thereby improving diversity in the language modality and reducing overfitting to fixed textual patterns.

Table A.1: Examples of generated sentence.

| Generated sentence |
| --- |
| The image shows a hand holding tench. |
| This is a close-up of a goldfish. |
| The image depicts a jay with its wings spread. |
| It would be a smooth newt with a smooth skin. |
| The peafowl is pecking at the ground. |
| The macaw is a vibrant and colorful bird. |
| The image features a Bluetick Coonhound. |

**Number of effective concepts (NEC).** NEC, originally proposed by Srivastava et al. (2024), is a metric that helps prevent information leakage by constraining model reliance on a limited set of semantically meaningful features. We adapt this approach to our multimodal CBM when computing the alignment score between an input image $x_i$ and its corresponding label $y_i$ using interpretable encodings. Specifically, we select the top-$n$ dimensions from the element-wise similarity between image and text encodings and use their sum as the final similarity score:

$$\text{logits} = \left( \frac{\sum \text{top-}n(I_e \odot T_e)}{\|I_e\|_2 \|T_e\|_2} \right) \times e^\tau \tag{A.8}$$

Here, $\odot$ denotes element-wise multiplication. Our method dynamically identifies the top-$n$ most relevant concepts for each image-text pair, making the reasoning process more interpretable and supporting better downstream interventions.

**Non-negative concept representation space.** In our concept representation space, each dimension reflects the similarity between the input (image or text) and a specific concept. To improve interpretability, we enforce a non-negative constraint on these activations by applying a ReLU function to both image and text embeddings: $I_e^+ = \text{ReLU}(I_e)$ and $T_e^+ = \text{ReLU}(T_e)$. This design improves interpretability in the following three aspects:

1. *Disambiguating negative responses.* As discussed by Sun et al. (2025), it is often unclear whether a negative activation implies the negation of a concept or its complete absence. By removing negative values, we avoid this ambiguity.

2. *Amplifying relevant concept activations.* Since similarity computations involve normalization, weak activations in high-dimensional spaces can lead to dilution of important signals. By zeroing out irrelevant (negative) dimensions, we strengthen the contribution of meaningful concepts. In the worst-case scenario, each dimension has a value of at most $\sqrt{\frac{1}{|C|}}$, where $|C|$ is the number of candidate concepts; thus, filtering noise is crucial.

3. *Improving inference reliability and efficiency.* Without non-negativity, the product of two negative activations (from image and text encodings) may yield a misleadingly high similarity score, falsely indicating semantic alignment. Enforcing non-negativity eliminates this issue and also simplifies the computation and sorting steps during inference.

Table A.2: Knowledge distillation accuracy comparison with black-box CLIP.

| Method | Dataset | | | | | | |
|---|---|---|---|---|---|---|---|
| | CIFAR-10 | CIFAR-100 | CUB | Food | OxfordPets | DTD | ImageNet |
| CLIP ViT-L/14 | 96.2 | 77.9 | 62.3 | 92.9 | 93.5 | 55.3 | 75.3 |
| **MM-CBM w/ KD** | 91.7 | 73.3 | 61.7 | 92.5 | 88.9 | 34.7 | 74.7 |

## A.5 Unsupervised setting via knowledge distillation

When ground-truth class labels are unavailable, we adopt the predictions of the backbone VLM (e.g., CLIP) as soft supervision. This unsupervised learning strategy enhances the flexibility of our framework, enabling the use of large-scale unlabeled images from the target domain together with only the labels of interest, thereby fully exploiting CLIP's representation capabilities.

Inspired by prior work on knowledge distillation (Hinton et al., 2015; Yang et al., 2024), we align the output distributions of our model with those of the VLM in both image-to-text and text-to-image directions. Let $M_{ij} = \cos((I_e)_i, (T_e)_j)$ denote the similarity matrix in the concept space, and $N_{ij} = \cos(E_I(x_i), E_T(y_j))$ the similarity matrix from CLIP. All embeddings are $L_2$-normalized. The corresponding softmax-normalized distributions are:

$$p_T = \text{softmax}(N), \; p_S = \text{softmax}(N^\top) \tag{A.9}$$

$$p_S = \text{softmax}(M), \; q_S = \text{softmax}(M^\top) \tag{A.10}$$

We then minimize the Kullback–Leibler (KL) divergence between the teacher (CLIP) and student (CBL) distributions:

$$L_{\text{KD}_{\text{I}\to\text{T}}} = D_{\text{KL}}(p_T \| p_S) = \sum_i p_T(i) \log \frac{p_T(i)}{p_S(i)}, \tag{A.11}$$

$$L_{\text{KD}_{\text{T}\to\text{I}}} = D_{\text{KL}}(q_T \| q_S) = \sum_i q_T(i) \log \frac{q_T(i)}{q_S(i)}, \tag{A.12}$$

$$L_{\text{KD}} = \frac{1}{2} \left( L_{\text{KD}_{\text{I}\to\text{T}}} + L_{\text{KD}_{\text{T}\to\text{I}}} \right). \tag{A.13}$$

Additionally, we treat CLIP's top-1 prediction as a pseudo-label $\hat{l}$ to supervise the classification head:

$$L_{\text{ACC}}^{\text{KD}} = L_{\text{CE}} \left( \frac{I_e \cdot T_e}{\|I_e\|_2 \|T_e\|_2} \cdot e^\tau, \; \hat{l} \right) + L_{\text{KD}}. \tag{A.14}$$

This unsupervised task-performance loss can be directly incorporated into the final objective in Equation 10, replacing the supervised loss, thereby enabling end-to-end training of an interpretable CLIP without requiring labeled data.

As shown in Table A.2, the knowledge-distilled MM-CBM largely preserves task performance across the other six datasets. In contrast, its performance on the DTD dataset is noticeably weaker. A plausible

Table A.3: Test accuracy comparison with CBMs on zero-shot settings. *Since LaBo (Yang et al., 2023) does not provide an official OxfordPets concept set, we conducted evaluation using our own data.

| Method | Dataset | | | | | | |
|---|---|---|---|---|---|---|---|
| | CIFAR-10 | CIFAR-100 | CUB | Food | OxfordPets | DTD | ImageNet |
| CLIP ViT-L/14 | 96.2 | 77.9 | 62.3 | 92.9 | 93.5 | 55.3 | 75.3 |
| LaBo (Yang et al., 2023) | 82.1 | 43.4 | 16.2 | 54.0 | 13.9* | 38.1 | 37.8 |
| ALBM (Zhang et al., 2025) | 83.1 | 43.1 | 25.0 | 75.4 | 35.9 | 48.5 | 64.6 |
| **MM-CBM(Ours)** | **95.3** | **75.7** | **36.3** | **84.3** | **76.6** | **50.2** | **66.1** |

explanation is that the black-box model itself performs poorly on DTD, resulting in soft labels that lack sufficiently informative latent knowledge, which in turn limits the effectiveness of the distilled model. This result demonstrates the strong scalability of our approach: given an image and an associated category of interest, it can achieve performance close to that of the black-box model, thereby greatly broadening the range of potential applications for MM-CBM.

## A.6 Supplementary experimental results

Tables A.4 and A.5 summarize the datasets and training configurations used in our experiments. Table A.4 reports the number of classes along with the train/test splits, where we follow the original dataset partitions. Table A.5 provides the dataset-specific hyperparameters, including batch size, number of training epochs, and the size of the concept set. Unless otherwise stated, we use a fixed trade-off weight of $\lambda = 0.2$, initialize the temperature at $\tau = 0.07$, and set the NEC threshold to 5 for all experiments.

Beyond these primary settings, we further compare MM-CBM with other models designed to improve generalization. As shown in Table A.3, MM-CBM achieves the strongest performance among all baselines.

Table A.4: Dataset Details.

| Dataset | Classes | Train size | Test size |
|---|---|---|---|
| CIFAR-10 | 10 | 50,000 | 10,000 |
| CIFAR-100 | 100 | 50,000 | 10,000 |
| CUB | 200 | 5,994 | 5,794 |
| Food | 101 | 75,750 | 25,250 |
| OxfordPets | 37 | 3,680 | 3,669 |
| DTD | 47 | 3,760 | 1,880 |
| ImageNet | 1000 | 1,281,167 | 50,000 |

Table A.5: Hyperparameter for each dataset.

| Dataset | Batch size | # of epochs | # of concepts |
|---|---|---|---|
| CIFAR-10 | 128 | 50 | 141 |
| CIFAR-100 | 64 | 50 | 795 |
| CUB | 8 | 50 | 604 |
| Food | 128 | 50 | 755 |
| OxfordPets | 8 | 50 | 205 |
| DTD | 4 | 50 | 365 |
| ImageNet | 256 | 12 | 4553 |

### A.7 Supplementary ablation studies

**Ablation study: number of effective concepts.** We examine the effect of the number of effective concepts (NEC) by comparing performance under NEC = 5 with using all concept activations for computing alignment scores. Specifically, we measure the ratio of the top five activations to the total contribution. As noted by Srivastava et al. (2024), CBMs trained with sparse concept supervision tend to rely on a small set of high-response concepts, improving robustness to NEC variations. Our results in Fig. A.1 exhibit the same tendency: even when all activations are used, the top two dominate the decision. Enforcing NEC = 5 further sharpens the activation distribution and increases variance, indicating that the model selects more distinct and informative concepts. This property can be leveraged to refine the concept set, producing explanations that are both concise and interpretable.

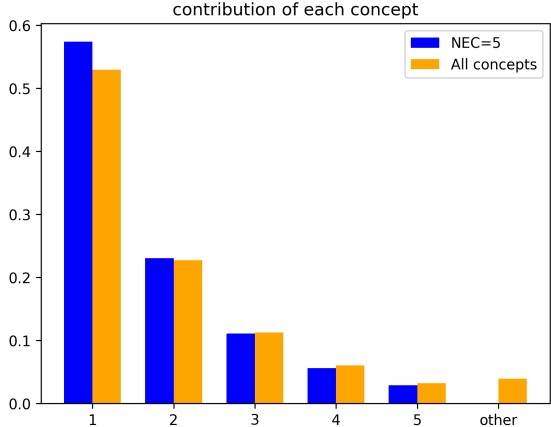

Figure A.1: Contribution of each concept used as explanation.

**Ablation study: trade-off weight $\lambda$.** We further analyze the influence of the trade-off weight $\lambda$ that balances task loss and interpretability loss. As illustrated in Fig. A.2, incorporating the interpretability loss consistently improves performance relative to optimizing the task loss alone. This effect likely arises because concept-based supervision provides richer semantic structure than raw image–label pairs, guiding the model toward human-aligned semantics and improving generalization. Remarkably, even when trained solely with the interpretability loss, MM-CBM still retains non-trivial task performance across four datasets, demonstrating the strong semantic learning capability induced by concept-level alignment.

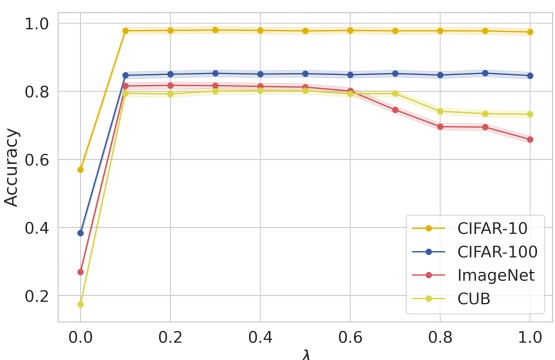

Figure A.2: Accuracy across different $\lambda$ value.

**Ablation study: non-negative concept representation space.** We evaluate the influence of enforcing non-negative concept activations by comparing alternative transformation functions to the ReLU baseline described in Section A.4. In particular, we include a squared activation function that guarantees non-negativity.

To understand the effect of these transformations, we analyze the cumulative distribution functions (CDFs) of activation values for visual activations, text activations, and final decision activations. As shown in Fig. A.3, ReLU produces the highest overall activation magnitudes. Text activations, supervised via text similarity, exhibit tighter concentration and lower variance, whereas visual activations, trained via one-hot labels, show higher dispersion, enhancing interpretability by producing clearer concept separation. Moreover, the dot-product operation effectively filters redundant textual information, allowing decision activations to retain higher variance and facilitating the selection of salient, task-relevant concepts.

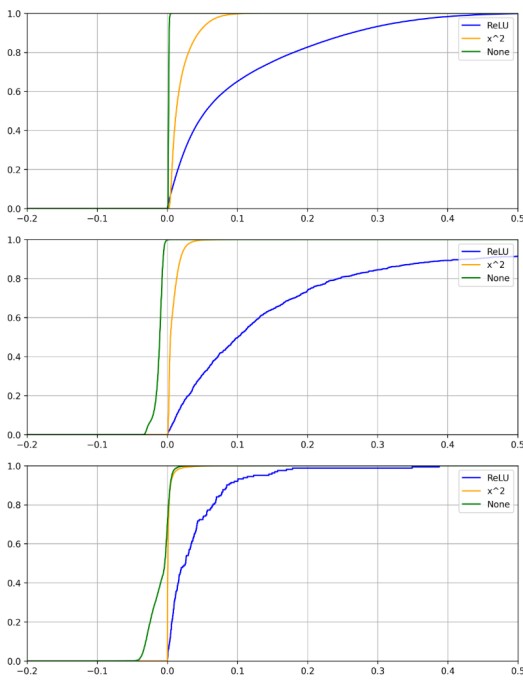

Figure A.3: Non-zero concept activation cumulative distribution function of final prediction(top), visual activation(middle) and language activation(bottom).

### A.8 Alternative Backbone

To evaluate the generalization capability of our approach, we conduct experiments with multiple variants of CLIP as well as other backbone models such as DINOv2. Since DINOv2 only provides an image encoder, we incorporate a SigLIP2 text encoder to facilitate the training process. The results in Table A.6 demonstrate that our method can be seamlessly adapted to different architectures with similar designs, rather than being restricted to interpretable variants of CLIP.

Table A.6: Accuracy comparison with stronger baseline under finetuned settings.

| Method | Dataset | | | | | | |
|---|---|---|---|---|---|---|---|
| Accuracy | CIFAR10 | CIFAR100 | CUB | DTD | Food | Pets | ImageNet |
| EVA-CLIP ViT-L/14 | 99.5 | 93.0 | 84.3 | 81.1 | 94.8 | 94.4 | 85.3 |
| **MM-CBM** | 99.3 | 92.7 | 82.8 | 82.0 | 94.3 | 93.6 | 83.8 |
| SigLIP ViT-L/16 | 97.8 | 87.4 | 84.4 | 82.8 | 95.4 | 94.9 | 86.4 |
| **MM-CBM** | 98.0 | 87.2 | 79.9 | 82.1 | 95.6 | 93.7 | 85.1 |
| SigLIP2 ViT-L/16 | 98.1 | 86.9 | 83.7 | 79.7 | 95.8 | 96.2 | 87.1 |
| **MM-CBM** | 98.2 | 86.4 | 77.8 | 82.8 | 95.9 | 93.9 | 85.7 |
| DINOv2 ViT-L/14 | 99.1 | 91.0 | 86.7 | 78.3 | 92.7 | 96.0 | 84.5 |
| **MM-CBM** | 99.2 | 91.2 | 84.6 | 78.0 | 93.0 | 94.5 | 83.1 |

## A.9 Image Retrieval

In Section 5, we define 5 different levels of queries and provide corresponding examples. In this section, we provide more cases and offer interpretable predictions in Fig. A.5 and Fig. A.4.

| | CLIP retrival | MM-CBM explainable results | | |
|---|---|---|---|---|
| concrete / Level-1 | Query: Give me an apple | | The concepts most related to query: | The concepts most related to the image |
| | | This image is an apple because it has:
1. *green apple (0.0259)*
2. *fruit (0.0211)*
3. *fruit basket (0.0133)* | apples (0.1479)
apple tree (0.1419)
red apple (0.1267)
apple (0.1110) | green apple (0.2572)
healthy food (0.2567)
human hand (0.2259)
fruit (0.2257) |
| Level-2 | Query: Metallic body with wheels | | silver metal (0.1056)
wheel (0.1017)
metal background (0.1015)
car wheel (0.1011)
alloy car (0.0994) | alloy wheels (0.3314)
parade (0.2326)
image (0.2193)
antique silver (0.2092)
bathroom mirror (0.2014) |
| | | This image contains a metallic body with wheels because it has:
1. *alloy wheels (0.0299)*
2. *antique silver (0.0171)*
3. *parade (0.0149)* | | |
| Level-3 | Query: Apple, the device | | apple logo (0.1438)
iphone (0.1246)
apple (0.1215)
ipod touch (0.1109)
apples (0.1107) | iphone (0.3094)
usb port (0.3005)
art glass (0.2589)
image (0.2535)
green screen (0.2426) |
| | | This image is an apple device because it has:
1. *iphone (0.0385)*
2. *usb port (0.0149)*
3. *green apple (0.0140)* | | |
| Level-4 | Query: Sharp mind | | human head (0.0841)
human brain (0.0743)
minds (0.0730)
brain (0.0715)
mindset (0.0689) | repair shop (0.3101)
coffee grinder (0.2269)
pen (0.2248)
male child (0.1955)
gun (0.1810) |
| | | This image contains a sharp mind because it has:
1. *human head (0.0121)*
2. *pen (0.0086)*
3. *creative business (0.0059)* | | |
| abstract / Level-5 | Query: Break a leg! | | theater production (0.1258)
leg match (0.0825)
dress rehearsal (0.0813)
straight leg (0.0766)
performs onstage (0.0746) | medical assistant (0.2252)
green screen (0.2063)
bones (0.2047)
car crash (0.1926)
lyrics (0.1795) |
| | | This shows theatrical slang because it has:
1. *bones (0.0145)*
2. *car crash (0.0139)*
3. *theater production (0.0128)* | | |

Figure A.4: Image Retrieval on five different types of queries (failure cases in Level-4 and Level-5).

Figure A.5: Image Retrieval on five different types of queries. The top of the module shows the query statement we use, and the right side shows the most relevant concepts.

