# OpenReview forum: "Multimodal Concept Bottleneck Models"
_TMLR — Under review for TMLR_

### Review · Reviewer_KbqP · 2026-07-09

**Summary Of Contributions:**

This paper proposes MM-CBM, a multimodal concept bottleneck model that maps both image and text embeddings into a shared concept space. Instead of using a fixed linear classifier on top of image-side concepts, the method learns dual concept bottleneck layers for image and text, and performs prediction through concept-level image-text similarity. This design extends CBMs toward CLIP-style zero-shot classification, retrieval, and free-text querying.

The main strengths are the clean formulation, the competitive ANEC-5 results against prior CBM baselines in Table 2, and the sparse concept-level explanations, which make predictions more inspectable than black-box CLIP. My main concerns are that several claims are stronger than the evidence currently supports, especially regarding preservation of CLIP performance, intervention, transfer/generalization, and the role of concept vocabulary size.

**Audience:**

Yes

**Audience Explanation:**

The paper addresses a topic of clear interest to parts of the TMLR audience: concept bottleneck models, interpretable vision-language models, and CLIP-style open-vocabulary prediction. The dual image/text concept bottleneck formulation is clean and may be useful for researchers working on concept-based interpretability or multimodal explanation.

Even though I have concerns about the strength of the empirical support, the direction itself is relevant. The results also provide useful evidence about the trade-off between CLIP performance and concept-level inspectability.

**Claims And Evidence:**

No

**Claims Explanation:**

Several claims are only partially supported by the reported evidence.

First, the paper claims in the abstract and Section 1 that MM-CBM remains within 5% of black-box performance. Table 3 supports this only in an averaged sense for the fine-tuned classification setting. Several individual datasets show larger drops: CUB drops from 84.5 to 75.0, and DTD drops from 82.1 to 75.6. In the zero-shot setting, the degradation is more substantial: CUB drops from 62.3 to 36.3, OxfordPets from 93.5 to 76.6, and ImageNet from 75.3 to 66.1.

The retrieval claim in Section 4.2 is also overstated. The paper states that the retrieval performance gap remains under 5%, while Table 4 shows R@1 gaps of 13.6 on Flickr30k text retrieval and 14.4 on MSCOCO text retrieval. These results may still be reasonable for an interpretable model, but they do not support the current wording.

Second, the intervention evidence in Section 4.4 is too limited. The main quantitative result is based on a 100-sample subset, where manually editing one text-side concept corrects 5 predictions and introduces 2 new errors. This is useful as an error-tracing example, but it does not convincingly demonstrate systematic human intervention or controllability.

Third, the zero-shot and transfer claims need stronger evidence. Section 4.1 and Table 3 should more clearly specify which data, labels, and concept vocabularies are used to train the CBLs in the zero-shot setting. The CUB results are especially concerning, with zero-shot accuracy dropping from 62.3 to 36.3 and fine-tuned accuracy dropping from 84.5 to 75.0. This suggests that concept coverage remains a bottleneck for fine-grained transfer.

Finally, the concept vocabulary size is an important uncontrolled factor. Table A.5 reports very different vocabulary sizes across datasets, including 4553 concepts for ImageNet, 795 for CIFAR-100, 604 for CUB, and 205 for OxfordPets. This size directly affects bottleneck capacity: larger concept dictionaries increase semantic coverage, increase the chance that discriminative attributes are available, and give the top-k scoring rule more candidate concepts to select from. It also affects interpretability, since large concept banks may contain redundant or spurious concepts that produce plausible-looking explanations. Appendix A.7 studies NEC, but this is different from ablating the total dictionary size.

Overall, the paper presents a reasonable and potentially useful method, but the current evidence does not fully support the strongest claims about preserving CLIP performance, enabling reliable intervention, and achieving robust open-vocabulary generalization.

**Requested Changes:**

Critical changes:
Tone down or precisely qualify the performance claims.
The claims about remaining within 5% of black-box performance and retrieval performance should be revised to match Table 3 and Table 4. If the claim refers to an averaged metric, the averaging protocol should be explicitly stated.

Add a more systematic intervention evaluation.
Section 4.4 should go beyond the current small case study. Please evaluate concept edits over a larger set of errors and report both correction rates and collateral damage, e.g., whether deleting or modifying a concept hurts other classes or previously correct predictions.

Clarify and strengthen the zero-shot/transfer evaluation.
Section 4.1 should clearly specify which data, labels, and concept vocabularies are used in the zero-shot setting. A direct cross-dataset transfer experiment would substantially strengthen the paper, e.g., training the CBLs on CC12M or one source dataset and evaluating on multiple unseen datasets with a fixed concept vocabulary.

Control for concept vocabulary size.
Please add an ablation over total concept dictionary size, not only NEC. For example, evaluate smaller/larger vocabularies or matched-size noisy/shuffled concept dictionaries. This is important because vocabulary size directly affects both predictive capacity and the apparent quality of sparse explanations.

Important but less critical changes:
Improve reproducibility details for concept generation.
Appendix A.3 reports an average pairwise Jaccard similarity of 0.48 across repeated concept generations on CIFAR-100. Please report decoding details such as temperature, top-p, seed, number of generated concepts per class, and few-shot exemplars.

Clarify baseline protocols.
Section 4.1, Table 2, Table 3, and Table A.3 should more clearly report backbone choices, NEC usage, concept sources, and concept vocabulary sizes across methods.

Clarify the interpretation of human/VLM preference studies.
The human and VLM-based evaluations in Section 4.5 are useful preference studies. Please avoid presenting them as strong evidence of faithfulness without additional causal perturbation or intervention experiments.

---

### Review · Reviewer_ZHyi · 2026-07-14

**Summary Of Contributions:**

This paper proposes MM-CBM, a dual concept bottleneck framework that extends CBMs to vision-language models, enabling interpretable open-vocabulary classification and retrieval while maintaining performance close to CLIP.

To train the model, the authors automatically construct concept annotations using LLM-generated concept vocabularies, OWLv2-based visual concept labels, and text–concept similarity scores. MM-CBM jointly optimizes concept alignment and task performance through dual concept bottleneck layers, producing concept-level explanations for both images and text.

Experiments on multiple image classification and retrieval benchmarks show that MM-CBM substantially outperforms prior CBM approaches while maintaining performance close to CLIP in both fine-tuned and zero-shot settings. The model also supports human intervention by modifying concept activations and provides interpretable explanations through the concepts most responsible for each prediction.

Strengths:

- Well motivated
- The method makes a lot of sense to me
- The results look promising

Weakness:

- Limited validation for the dataset used for alignments. The proposed framework relies heavily on automatically generated concepts and annotations from multiple pretrained models (LLMs, OWLv2),  yet provides limited evidence that these concepts faithfully align with human-understandable semantics. Since interpretability is a central claim of the paper, additional validation of concept quality and reliability would strengthen the credibility of the proposed explanations.

- Modality imbalance. The paper demonstrates strong performance on object- and attribute-centric tasks, but its own case studies reveal challenges on more abstract or semantic queries. Since visual concepts are derived from object-detection supervision, the learned concept space appears biased toward concrete objects, which may limit the framework’s ability to support broader open-vocabulary semantic reasoning.

**Audience:**

Yes

**Audience Explanation:**

N/A

**Claims And Evidence:**

Yes

**Claims Explanation:**

N/A

**Requested Changes:**

- Since the interpretability claims heavily depend on the quality of the generated concepts, could the authors provide evidence or a systematic validation protocol demonstrating that the automatically generated concepts and annotations are reliable and aligned with human-understandable semantics?

- The visual concepts are generated from OWLv2 object detection, which primarily captures object-level attributes. Why did the authors choose object-detection-based supervision instead of leveraging VLM-generated semantic concepts? Could VLM-based concept annotation help alleviate the modality imbalance observed in abstract semantic queries?

---

### Review · Reviewer_r7xk · 2026-07-20

**Summary Of Contributions:**

This paper proposes Multimodal Concept Bottleneck Models (MM-CBM), which extend concept bottleneck models to vision-language architectures. MM-CBM uses separate image and text concept bottleneck layers to project both modalities into a shared, named concept space. Predictions are computed directly from non-negative image-text concept activations rather than through a learned linear classification head. In the supervised setting, the score is based on the top-$n$ element-wise concept products, normalized by the full image and text concept-vector norms.

The supervision pipeline is largely automated. Candidate concepts are generated by an LLM, visual concept targets are produced using OWLv2, and text-side concept targets are based on sentence-embedding similarity. The paper evaluates classification, zero-shot transfer, image–text retrieval, concept interventions, human and VLM explanation preferences, knowledge distillation, and compatibility with several alternative backbones.

The main strengths are the simple dual-bottleneck construction, removal of the fixed post-concept classifier, support for free-text queries, broad experimental coverage, and competitive performance against prior CBMs. In Table 2, MM-CBM obtains the highest reported average ANEC-5 accuracy, $71.08$, compared with $68.60$ for VLG-CBM. The six reported image-retrieval gaps relative to CLIP in Table 4 are all below five percentage points. Appendix Table A.6 also provides useful evidence that the construction can be applied to several backbone families.

The principal weakness is that the paper's strongest claims, such as “fully transparent inference,” “complete interpretability,” and “more faithful explanations”, are stronger than the evidence presented. The scores are computed in a named concept space, but the semantic validity and faithfulness of the learned coordinates are not independently evaluated. Several numerical claims also require more precise wording, the human-study protocol is under-specified, the visual concept-target construction is ambiguous, and some experimental and mathematical details require clarification.

**Audience:**

Yes

**Audience Explanation:**

The paper addresses a relevant intersection of concept bottleneck models, vision-language models, open-vocabulary recognition, interpretable classification, and image-text retrieval. The dual image/text bottleneck design is simple and potentially useful, especially because it removes the fixed post-concept classification head and supports textual queries through the same named concept basis as images.

The results should be of interest to researchers studying interpretable multimodal models. In particular, the paper reports strong average performance against prior CBMs, relatively small image-retrieval degradation compared with CLIP, meaningful limitations on fine-grained and abstract queries, an example of concept-level intervention, and compatibility with several alternative backbones. These findings remain useful even if the strongest interpretability claims require revision.

**Claims And Evidence:**

No

**Claims Explanation:**

The evidence supports MM-CBM's competitive performance against the reported CBM baselines and participants' preference for its explanations, but it does not fully support the stronger claims of complete interpretability, fully transparent reasoning, or faithful explanations:

1. **Concept semantics are not independently validated.** Although predictions are computed from named concept-layer outputs, the paper does not independently evaluate whether the learned coordinates consistently possess their assigned semantic meanings, for example through concept-level annotations or a systematic audit. The evidence supports an inspectable named concept space, but not the stronger claim of complete semantic interpretability.

2. **Visual target construction is unclear.** The visual target and BCE loss are defined over the complete concept set $C$, but OWLv2 is queried only with the class-specific subset $C_{y_i}$. The paper should specify whether unqueried dimensions are masked from the loss, treated as unknown, or assigned zero. If they are assigned zero, visually present but unqueried concepts may be treated as absent, which would affect the semantic interpretation of the visual bottleneck.

3. **The displayed concept contributions need clarification.** Equation 8 computes the score from top-$n$ concept products with full-vector normalization, but the manuscript does not define precisely how the per-concept weights displayed in the explanation figures are obtained. The authors should state whether these weights are raw products or normalized contributions and whether the displayed contributions sum to the final similarity score.

4. **Experimental terminology should be more precise.** A CLIP linear probe should not be described interchangeably with full model fine-tuning. The zero-shot protocol should also be defined as using no target-dataset training examples or target-specific parameter fitting, rather than stating that the model has observed no data or labels at all. The model is trained on CC12M, and target class names may still be supplied as text queries at inference.

**Requested Changes:**

1. Substantiate or narrow the interpretability and faithfulness claims
Provide direct evidence that the learned concept coordinates consistently represent their assigned semantic meanings, or narrow claims such as “fully transparent inference,” “complete interpretability,” “renders CLIP interpretable,” and “more faithful explanations” to match the evidence presented. Appropriate validation could include independent concept-level evaluation, systematic human auditing of concept activations, or faithfulness-oriented sufficiency and comprehensiveness tests.

2. Clarify the visual concept-target construction
Specify how concept dimensions outside the class-specific queried set $C_{y_i}$ are handled in the visual BCE loss, including whether they are masked, treated as unknown, or assigned zero. If unqueried dimensions are assigned zero, evaluate whether this introduces false-negative targets for visually present but unqueried concepts, and mitigate the issue if necessary.

3. Clarify the training, comparison, and scoring procedures
For each experiment, specify which encoders and bottleneck layers are frozen or trained. Distinguish clearly between linear probing and full fine-tuning. Define the zero-shot setting as using no target-dataset training examples or target-specific parameter fitting, and clarify how target class names are used at inference. Also explain how the per-concept weights displayed in the figures are derived from Equation 8. State whether they are raw concept products, normalized contributions, or otherwise transformed, and whether the displayed contributions sum to the reported final score.